# Differences in Drug Use among Persons Experiencing Homelessness According to Gender and Nationality

**DOI:** 10.3390/ijerph20054007

**Published:** 2023-02-23

**Authors:** Alícia Parés-Bayerri, Fran Calvo, Sílvia Font-Mayolas, Sonia Panadero, José Juan Vázquez

**Affiliations:** 1Social Psychology Department, Universitat de Girona, 17003 Girona, Spain; 2Serra Húnter Fellow, Pedagogy Department, Quality of Life Research Institute, Universitat de Girona, Plaça de Sant Domènec, 17003 Girona, Spain; 3Psychology Department, Quality of Life Research Institute, Universitat de Girona, 17003 Girona, Spain; 4Clinical Psychology Department, Universidad Complutense de Madrid, 28040 Madrid, Spain; 5Social Psychology Department, Universidad de Alcalá, 28871 Madrid, Spain

**Keywords:** Spain, homelessness, substance use disorders, drug abuse, addiction, immigrants, mental health

## Abstract

The main aims of this article are to update the data related to drug and alcohol use in persons experiencing homelessness (PEH) who use shelters, and to see if there are significant differences in their drug use depending on their gender and nationality. The article presents an analysis of the interconnections between the results of drug dependence detection tools (Alcohol Use Disorders Identification Test (AUDIT), Drug Abuse Screening Test (DAST-10), Severity of Dependence Scale (SDS)) according to gender and nationality with the intention of identifying specific needs that lead to new lines of research into better approaches to homelessness. A cross-sectional, observational and analytical method was used to analyse the experiences of persons experiencing homelessness who use various shelters in the cities of Madrid, Girona, and Guadalajara (Spain). The results show that there are no gender differences in the risks of using drugs and drug addiction, but there are differences in terms of nationality for drug addiction, with Spanish nationals showing a greater tendency to develop drug addiction. These findings have significant implications, as they highlight socio-cultural and socio-educational influence as risk factors in drug addiction behaviours.

## 1. Introduction

Homelessness is increasingly prevalent in cities in the developed world. The United Nations Commission on Human Rights estimates that there are 100 million persons experiencing homelessness (PEH) in the world and 3 million in Europe. In Spain, between 2014 and 2016, the number of people using shelters increased by 20.5% on average, reaching 16,437 PEH in Spain in 2016 [1]. The Instituto Nacional de Estadística de España (INE) (the Spanish National Institute of Statistics) registered 17,772 PEH in day centres and shelters during 2020 [2] and considers that a total of 28,552 PEH live in Spain [3].

The European Federation of National Organisations Working with the Homeless bases their concept of residential exclusion on the European Typology of Homelessness and Housing Exclusion (ETHOS) categories. It defines PEH as those who cannot access or maintain adequate and permanent personal accommodation that provides them with a stable living environment, either due to a lack of economic resources or because they have personal or social difficulties in leading an autonomous life [4].

Homelessness is not static; it is the result of a process of changes in life that lead a person to being in a situation of exclusion. Causal factors of homelessness are structural, such as unemployment and economic crises [5,6], and individual, such as nationality, gender, or use of drugs [7]. Both occur within the individual’s circle, resulting in inequalities in rights, freedoms, opportunities, and personal capabilities [8]. Loneliness is one of these inequalities [9]; suffering violence of all kinds is another [10,11]. The process of social exclusion leads to PEH being alienated from social participation [12].

The risk factors affecting PEH are varied. Although mental health and addiction stand out, interventions cannot be implemented and homelessness cannot be addressed uniformly, but must be based on the different realities of each PEH [13].

Drug use by PEH has been studied and reported in the scientific literature. The conclusions are that PEH are more vulnerable than the rest of the population and have a higher propensity to developing drug addiction. Alcohol is the most commonly consumed substance [14,15]. PEH are much more likely to be dependent on alcohol and other drugs than their peers of the same age group in the general population [16].

Similar research shows that between 30% and 70% of PEH abuse alcohol or drugs, triggering disorders associated with drug use that affect almost 50% of PEH in Spain, such as health problems, psychiatric symptoms, and poly-drug use [17].

In their research analysing substance use according to the amount of time spent homeless, [18] found that alcohol use was much more common among those who had been homeless for over 5 years than in the group that had been homeless for less time. Longer-term PEH suffer from severe physical and mental health problems, and alcohol and other drug use: 25% to 45% of PEH have a disorder caused by alcohol abuse [19] and 53% of PEH will have some form of addiction during their lifetime [20]. Fajardo’s 2011 study on addictions in PEH stated that the most common addictions were alcohol (43.48%), heroin (13.04%), pathological gambling (13.04%), a mix of heroin and cocaine (8.70%), and cocaine (4.35%). The most commonly diagnosed disorders among PEH were those related to the use of alcohol, 36.7% (95% CI 27.7% to 46.2%), those related to the use of drugs, 21.7% (95% CI 13.1% to 31.7%), followed by schizophrenia spectrum disorders 12.4% (95% CI 9.5% to 15.7%), and clinical depression 12.6% (95% CI 8.0% to 18.2%) [21].

PEH are subjected to discriminatory treatment resulting from unconscious biases, prejudices, and stigmas that are anchored to a social imaginary [22], and when we refer to homeless women, a double stigmatisation is observed: on the one hand, inequality and social discrimination due to the fact of being a woman, and on the other, the negative opinion entailed in being a homeless person, since this scenario opposes the classic female role [23,24]. This double stigmatisation also occurs in immigrants, the stigma entailed in being homeless being shared with separation from the rest of society due to the fact of being foreign, thus making it even more difficult to establish bridges between the two “worlds” [25].

The aim of this research is to determine the real situation with regard to drug dependence among homeless women and immigrants, these two groups being prone to greater stigmatisation than those of men and nationals, respectively.

### 1.1. Substance Use and Homeless Women

The application of research models in the social field whose dimensions do not incorporate the gender perspective has resulted in women traditionally being invisible in studies on homelessness [26]. This article has taken the above issue into consideration, and therefore offers results on drug dependence in homeless people differentiating by gender.

Women experiencing homelessness are more vulnerable than men in terms of health issues, as they suffer from more sexually transmitted, circulatory, muscular, skeletal, and dermatological diseases than men [27,28]. In addition, many of these women have experienced severe and violent assault [29], which exacerbates homelessness and increases the likelihood of suffering a more severe mental health disorder [30].

Most of the women who use drugs were drug users before they lost their homes; in some cases, drug use was the cause of them losing their home. Their drug use intensifies considerably when they become homeless, leading to other health problems. There is also a group of women who start using drugs when they become homeless [31]. Diseases associated with drug use, such as hepatitis C and AIDS, are more prevalent in homeless women than in men [29].

Other studies have shown that drug and alcohol use among homeless women is lower than among men. As in the rest of the population, this is due to patriarchal factors that normalise consumption in men and hide it in women. Women, however, are addicted for longer, causing greater deterioration [32].

As well as being a woman, another factor that has a direct impact on homelessness in Spain is not having Spanish nationality [33].

There is a gap in the literature when it comes to studies that compare drug use by gender in homeless people. Although one U.S. study comparing behavioural factors of alcohol consumption between homeless men and women concluded that no differences could be detected in various manifestations of problematic alcohol use [34], the negative meta-stereotypes associated with homeless people (for example, alcohol users, drug users, slackers, delinquents, etc.) are more attributed to women than men [35], and even though homelessness affects both men and women, it is the latter who are most affected, since homeless women suffer the double discrimination of gender and social class, which sometimes intersect with other exclusion factors [36].

### 1.2. Substance Use and Homeless Immigration

In recent years, there has been an exponential increase in the number of immigrants living in Spain. The INE (2022) states that of the 28,552 PEH residing in Spain, 28.8% of them became homeless due to the need to start from scratch after arriving from another country.

The majority of immigrant PEH are from Africa (53.3%), followed by America (25.9%), and Europe (16.7%). In total, 43% of these immigrant PEH have been in Spain for over five years.

Immigrant homeless women are a more vulnerable subpopulation within this context. Many of them do not have information about their rights to access medical services in the country where they live. This situation is compounded by bureaucratic procedures, as well as many other obstacles, such as discrimination based on race, language, and cultural barriers [37,38].

In terms of health, they have great problems in accessing health care services [39,40]. In addition to the lack of documentation, they report psychological barriers, such as embarrassment about appearance and hygiene, and environmental barriers, such as a lack of knowledge regarding how the health care services work. Waiting times to be seen by a physician are long, which means they often give up seeking medical care [41]. In many cases, they also do not go to health care services for fear of being discovered and reported [42,43].

Studies on immigrant populations show that they suffer more distress than local-born people [44,45]. One of the reasons is the process of gaining residence in Western countries [46].

Paradoxically, there is a phenomenon in immigrant PEH called the healthy immigrant effect (HIE) that is related to the positive relationship between migration and health; immigrant PEH have better health than the local population of the same age [47,48,49]. Specifically, immigrant PEH have better overall health status, better mental health status, and less problematic substance use [50,51,52].

In Spain, a study was carried out in Bilbao on 107 non-Spanish PEH: 97.25% were men, 65% were from North Africa, 21.7% from Sub-Saharan Africa, 8.3% from Latin America, 3.3% from Eastern Europe, and 1.7% from Asia. A large proportion (76.7%) had no documentation. In total, 40% consumed alcohol on a regular basis [33].

This study aims to analyse whether there are differences in drug use between male and female PEH and between those born locally and immigrants.

Given that men and women differ in their biotype, physiology, and relationships with their environment, it was expected that different levels of drugs consumption would result in differences in abuse and dependence between the two genders. In addition, we also wanted to verify whether the concept of healthy immigration also occurs in the homeless immigrant population, despite all the difficulties of living on the street being combined with having undergone a very stressful migratory process.

The aims of this research are:(1)To show whether there are any differences in the level of drug dependence between homeless men and women (according to addiction screening tools).(2)To show whether there are differences in the level of drug dependence between homeless immigrants and nationals (according to addiction screening tools).(3)To explain the main consequences of such differences and the inequalities they generate for both comparisons.(4)To provide a social explanation for these differences.

The main questions that arose from the challenge of this research were as follows:(1)Are there differences in the consumption and degree of addiction to drugs in homeless men or women?(2)Are there differences in the consumption and degree of drug addiction in homeless immigrants and nationals?(3)What conclusions can be drawn to improve the planning and execution of care and prevention actions for drug addiction in homeless people?

Determining the level of drug dependency presented by homeless women and homeless immigrants in comparison with nationals will provide information on the health of the former two groups, allowing actions to be designated that meet their specific needs.

## 2. Materials and Methods

### 2.1. Design

Cross-sectional, observational and analytical study.

### 2.2. Participants

The study was based on a sample of 325 PEH interviewed in Madrid (*n* = 259), Girona (*n* = 51), and Guadalajara (*n* = 15). The participants were of legal age. The inclusion criterion was belonging to operational categories 1, 2, or 3 (most severe homelessness) of the ETHOS classification [4] at the time of the interview. See Table 1.

### 2.3. Procedure

This study was based on data provided by people interviewed in various cities:Women experiencing homelessness in Madrid staying overnight in accommodation for PEH (35.3%, *n* = 115). Given the relatively limited number of women experiencing homelessness in Madrid who were staying overnight in shelters or accommodation facilities for PEH, all the women who agreed to participate and met the criterion of ETHOS operational categories 2 and 3 were interviewed. A total of 6 transgender women were added to the description of the sample.Men experiencing homelessness in Madrid staying overnight in accommodation for PEH (42.4%, *n* = 138). A representative sample of men experiencing homelessness who met the criteria of ETHOS operational categories 2 and 3. A random sampling strategy was used in all accommodation for men experiencing homelessness in Madrid; the number of participants was chosen proportionally and randomly in each care facility according to its capacity.People experiencing homelessness in Guadalajara (4.6%, *n* = 15). All PEH in Guadalajara (men 66.7%, *n* = 10; women 33.3%, *n* = 5) who met the criteria of ETHOS operational categories 1 and 2 and agreed to be interviewed.People experiencing homelessness in Girona (15.6%, *n* = 51). PEH (men 76.4%, *n* = 39; women 19.6%, *n* = 10; people who did not wish to express their gender identity 3.2%, *n* = 2) who met the criterion of ETHOS operational categories 2 and 3 and agreed to be interviewed.

The purpose of the research and the way the data would be used were explained to all the participants. Their informed consent to being interviewed was requested, with the guarantee that their anonymity would be respected at all times. A structured interview was used for the collection of information, helping to overcome any possible reading and/or comprehension difficulties. The structured interviews lasted between 45 and 80 min.

### 2.4. Instruments

As a measuring instrument, a structured interview was created for collecting data in various areas. Socio-demographic and drug use data were collected for this study.

Socio-demographic variables included gender, age, nationality, marital status, sexual orientation, and level of education.

The following standardised instruments were used to detect the level of substance dependence.

The Alcohol Use Disorders Identification Test (AUDIT) [53], which assesses a person’s level of alcohol dependence. It is divided into two distinct parts:(a)A 10-question questionnaire about alcohol consumption in the previous year.(b)A medical examination and trauma history aimed at overcoming possible defensive attitudes of the user.

The Spanish version of the questionnaire [54] was used. Scores between 0 and 5 indicate abstinence from alcohol consumption, while scores of 6 or more indicate risky drinking.

The Drug Abuse Screening Test (DAST-10) [55] assesses problematic drug use for the following drugs: cocaine, heroin, methadone, cannabis, and sedatives. It also offers the option of including other drugs that the respondent uses. The Spanish validation of the instrument was used [56]; it has the correct psychometric properties. It establishes four levels: 0 No problems reported; 1–2 Low level; 3–5 Moderate level; 6–8 Substantial level; 9–10 Severe level.

The Severity of Dependence Scale (SDS) [57] was used to screen for drug dependence and severity. The SDS consists of five items that quantify the effect of drug use; each item has a Likert-type response from zero to four. The Spanish adaptation used [58,59,60] has the correct psychometric properties.

### 2.5. Statistical Analysis

Central tendency and dispersion measures were used to describe quantitative variables, while absolute and relative frequencies were used for categorical variables. The Kolmogorov–Smirnov test was used to determine the normality of quantitative scales. The Mann–Whitney U test was used to compare non-parametric independent samples and the chi-square test was used to compare categorical variables. All variables were analysed with a 95% confidence interval (CI) and an alpha level of significance of 0.05. IBM^®^ SPSS^®^ for Mac (version 22) software was used to analyse the data.

## 3. Results

### 3.1. Sociodemographic Data

Of the sample of 325 people, 57.5% were cisgender men, 40% cisgender women, and 1.8% transgender. The average age was 51.2 years (SD = 11.4). A total of 52.6% of the participants were Spanish, 40.6% were immigrants without Spanish nationality, and 6.5% had dual nationality. Single people accounted for 61.8% of the sample, followed by separated people (27.7%), married people (7.7%), and widows (3.4%). A total of 91.7% identified as heterosexual, 3.7% bisexual, 3.1% homosexual, and 0.9% identified as being in other categories.

Similar percentages of participants had a school leaving certificate (32.9%) to those with secondary education or vocational training (37%), followed by 16% with higher education, 8.6% with incomplete primary education, and 5.2% with no education (Table 2).

### 3.2. Drug Use

The mean DAST-10 score for drug use was 1.26 (SD = 2.64); 70.2% of the participating PEH showed low risk or abstinence, 3.1% risky drug use, 8% harmful use, and 9.8% dependence on drugs. Grouping these DAST-10 results together, 73.2% showed non-harmful drug use and 17.8% harmful use.

The SDS tool gave a mean score of 5.34 (SD = 4.86); 10.8% of PEH showed no drug dependency and 13.2% showed dependency.

Finally, the mean from the AUDIT tool was 12.37 (SD = 9.11), where 9.5% had no-risk use, 17.5% had risky use, and 14.5% had dependence. Grouping the scales results in 32.3% of the participating PEH showing harmful or risky drug use, and 26.5% showing drug dependence (Table 3).

The Kolmogorov–Smirnov test for a sample determined that the quantitative variables of the scales used to assess drug addiction were not normal (DAST-10 scores = 0.456, *p* < 0.000, SDS scores = 0.156, *p* < 0.001 and AUDIT scores = 0.136, *p* < 0.001).

### 3.3. Comparison of Drug Use by Gender

The mean score on the DAST-10 scale for men was 1.13, while for women it was 1.04. No statistically significant differences were found between men and women (*p* = 0.966), neither in the interpretation of this scale for low risk or abstinence (men = 80.1%, women = 79.8%), for risky use (men = 1.5%, women = 4.4%), or for harmful use (men = 8.8%, women = 7.0%) (*p* = 0.563). There were also no significant differences for dependence according to the DAST-10 tool (men = 9.6%, women 8.8%, *p* = 0.830). Grouping results within the DAST-10 scale, similar percentages were observed in non-harmful use (men = 81.6%, women = 84.2%) and in harmful use (men = 18.4%, women = 15.4%) (*p* = 0.588).

The mean SDS score for men was 5.74, and for women it was 5.22. No statistically significant differences were found between men and women (*p* = 0.668). There were also no differences in the interpretation of this scale for non-dependence (men = 48.1%, women = 50.0%), and for dependence (men = 51.9%, women = 50.0%) (*p* = 0.895).

The results of the AUDIT alcohol assessment tool showed a mean score of 11.47 for men and 14.73 for women (*p* = 0.225), so no significant differences were found. There were also no differences in the interpretation of this scale for non-risky drinking (men = 41.2%, women = 33.3%) or risky drinking (men = 58.8%, women = 66.7%) (*p* = 0.507). There was no difference in dependence (men = 50.0%, women = 63.6%) (*p* = 0.026). Grouping the scores confirmed the few differences there were: in harmful or risky use (men = 20.0%, women = 28.9%; *p* = 0.992) and dependence (men = 22.5%, women = 26.4%; *p* = 0.456) (Table 4).

### 3.4. Comparison of Drug Use by Nationality

The mean score of the DAST-10 scale for Spanish people was 1.63, and for immigrants it was 0.95. No statistically significant differences were found between the two (*p* = 0.088). There were no statistically significant differences in the interpretation of this scale for low risk or abstinence (Spanish = 72.4%, immigrants = 80.7%), for risky use (Spanish = 3.8%, immigrants = 3.4%), or for harmful use (Spanish = 8.3%, immigrants = 9.2%) (*p* = 0.167). According to DAST-10, however, there were significant differences for dependence (Spanish = 15.4%, immigrants = 6.7%, *p* = 0.026). When grouping the scores, differences in the percentages were observed in non-harmful use (Spanish = 76.3%, immigrants = 100%) and in harmful use (Spanish = 23.7%, immigrants = 19%) (*p* = 0.114).

The mean score on the SDS scale for Spanish people was 6.0, while for immigrants it was 4.66. Statistical differences were found between the two (*p* = 0.279). Statistical differences were also found in the interpretation of this scale for non-dependence (Spanish = 44.0%, immigrants = 40.0%) and for dependence (Spanish = 56.0%, immigrants = 60.0%) (*p* = 0.714).

The results of the AUDIT alcohol assessment tool showed a mean of 13.36 for Spanish people and 9.61 for immigrants (*p* = 0.265), so there were no significant differences between the two. There were differences in the interpretation of this scale for non-risky drinking (Spanish = 32.8%, immigrants = 39.1%) and risky drinking (Spanish = 67.2%, immigrants = 60.9%) (*p* = 0.586). There were also differences in dependence (Spanish = 55.7%, immigrants = 47.8%) (*p* = 0.057).

The pooling of results from all the scales (DAST-10, SDS and AUDIT) showed that there were significant differences in harmful or risky use (Spanish = 42.4%, immigrants = 22.0%, *p* = <0.001) and dependence (Spanish = 35.3%, immigrants = 18.2%, *p* = 0.001) (Table 5).

## 4. Discussion

The aim of this study was to analyse the differences in drug use and drug dependence in a sample of PEH according to their gender and nationality. The PEH who participated were users of various shelters in Spain. The study used DAST-10, SDS and AUDIT dependence assessment tools to compare men and women experiencing homelessness, and those born in Spain and those who immigrated there. Drug use was similar in men and women. There were more differences in terms of nationality, with immigrants showing less risky use than Spanish people.

An analysis of the socio-demographic variables of the sample shows that the average age was 51, slightly above that of the largest group in the latest INE press release [3], where 51.1% of PEH were under 45 and 43.3% between 45 and 64 years of age. Other studies in Spain [61,62,63] and internationally [64,65,66] established an average age for PEH of between 37 and 45. This research also confirmed that most PEH are single, as was the case in our study, where the percentage of single people was 61.8%.

A total of 37% of the sample of PEH had been in secondary education and 32.9% had a school-leaving certificate, which is proportional to the 2022 data provided by the INE. The latter show that 65.0% of PEH had been in secondary education, 23.8% had been in primary education or lower, and 11.3% had been in higher education [3].

### 4.1. Drug Use by Gender

Although the percentage of men (57.5%) is higher than that of women (40%), the representation of the latter is considerable if we bear in mind that research on drug use has historically ignored gender differences both in the data and the results, thus generating an under-representation of the realities of women, and, consequently, a biased analysis of the drug use problem being studied [67].

The literature on the comparison of the genders is varied. On the one hand, studies such as [68,69,70] state that there are no differences in drug use between men and women; although there are differences in variables, such as age of onset of use or relapse. On the other hand, there are studies that do show important differences, such as that of Allen [71], which shows that drug-dependent women suffer the side effects of drugs much more than men. Or the study in [72], where men appear to be a driving factor in women’s drug use, as many women start using drugs because their partners use them. Another characteristic that differentiates women’s drug use is that many of them also suffer from other disorders, so there is a large percentage who suffer from dual pathologies [73,74].

This study shows that there are no significant differences between men and women in their levels of drug use. Internationally, at the beginning of the 1990s, the Council of Europe reported an increase in drug use by women, who were rapidly moving towards reaching the same levels as men [75]. Spanish women have been found to be regular users of legal drugs; they consume more psychotropic drugs, tobacco, and alcohol than men [76].

In some studies, comparative data on men and women’s drug use come from quantitative databases stored by health centres, the latter being known to the health centres because women have expressly sought medical treatment. The results are, therefore, sometimes biased, as they are extracted from the medical records of women who are in poorer health already linked to community health resources [29].

The similar use of alcohol between men and women is deserving of analysis, because it can be misleading. Those responsible for public policies aimed at carrying out actions to improve the quality of life of the homeless might understand that no significant differences in drug dependence means that both genders should receive the same interventions. We, therefore, see the introduction of the gender perspective in future projects as being of special importance, because, although no differences are detected at a quantitative level, these data must be complemented with other qualitative data that contextualise and provide a more in-depth understanding of social and individual aspects relating to women. In the Introduction, we referred to the greater stigmatisation of homeless women; in addition to this, however, they are more vulnerable than men to the effects of drugs at an organic level, so the same consumption is actually much more harmful than in men, while they are also found to have greater difficulties in starting treatment for substance addiction than men. What is more, even when their resistance to asking for help is overcome, the evolution of their addiction is usually worse than that of men. The causes that lead to consumption, as well as the biological, psychological, and social impact of drugs, are also different between the sexes [77].

To give continuity to the results obtained, a future line of research could be to generate a discussion with a focus group of the women interviewed addressing issues such as motivation to consume, feelings of guilt for accessing treatments, gender roles, perceived stigma and self-stigmatisation, and mental and organic health. Public policies must incorporate the social context as a determining factor in differences in consumption between homeless men and women. We cannot talk about women and their actions or feelings as a homogeneous group, because the social and cultural moment in which they grow up and live also determines their behaviours, attitudes, and emotions. In addition, they must take into account differential aspects, such as, for example, that women identify affective factors as the cause of the onset of drug use and addiction. The influence of the partner and the lack of affectivity in the family environment are the most common causes cited by women themselves as factors that promote the initiation of consumption. However, these factors do not appear in any cases when it comes to men, where others emerge associated with personality (insecurity, shyness, or inability to socialise) or other contextual factors, such as work or the social environment (friends) [78].

The results of this study are based on structured interviews conducted individually and in a setting of intimacy and trust. This encourages women, whether or not they are linked to the mental health system, to be more open in telling their stories and expressing their drug use more frankly. This may explain the minimal differences between the genders. Women are not less addicted to drugs, but they talk less about their addiction and their reasons for drug use are different.

### 4.2. Drug Use by Nationality

Immigrants represented 47.1% of the sample. The main objective of those without citizenship was to complete the paperwork necessary to obtain it. When they first arrived in Spain, they sought to make use of the resources that are available for this. A lack of proper documentation places a person in a situation of total vulnerability. It prevents them from carrying out basic aspects of daily life, such as registering with the local council or accessing social benefits. A significant correlation has been shown between this and variables such as emotional distress or the length of time spent being homeless [79]. The factor that most influences immigrants’ extreme exclusion is a lack of the correct documentation; documentation that enables them to access work and social resources under the same conditions as Spanish people [80,81].

In this study, immigrant PEH used less drugs than Spanish PEH; they, therefore, had better health. The healthy immigrant effect (HIE) described in the scientific literature helps explain the health status of immigrants on arrival in Spain. The HIE says that immigrants who have recently arrived in a country have better health than the local population with similar socio-demographic characteristics. Their health eventually converges on the level of the local population as the number of years of residence increases [82]. The less time migrants have been in a country, the more differences in drug use will exist with local PEH. Different studies show a tendency for immigrants from developing societies to have healthier lifestyles than the local population, thus also having higher risk factors for harmful habits, such as tobacco and alcohol consumption [83,84,85].

Considering that the length of residence in a country of immigrant PEH is a significant determinant of their health status and, consequently, of an evolution towards poorer health that also involves drug use, health policies should take immigration into account, developing interventions that aim to preserve the HIE as long as possible and addressing immigrants’ changing needs.

Some limitations of this study should be considered. Firstly, the data came from structured interviews about drug use. Due to the stigma attached to PEH who use drugs, it is natural that some PEH did not answer some questions fully, so it is impossible for us to know objectively their real drug use. Secondly, 1.8% of the sample were transgender PEH, but data on them have not been taken into account because they are not conclusive and the percentage of the sample they represent is too small. Other studies have considered as a limitation not being able to include people of other genders in their comparisons, due, in part, to the way the data are collected [29]. Thirdly, the sample mainly included PEH from Madrid in ETHOS categories 2 and 3. There were much smaller samples from Girona and Guadalajara. The latter only included ETHOS categories 1 and 2. Therefore, the results mainly reflect the consumption of PEH in the Madrid. For future research, it would be interesting to include a proportional sample from other cities. It would also be interesting to investigate whether living in one place or another affects the level of drug use. Fourth, the low representativeness of the data in some results causes them to lose robustness. This is what happened in the case of the sample of five women living on the street in Guadalajara. Future research could aim to design qualitative work with these women. Fifthly, analyses were not carried out according to gender and immigration, taking into account the low sample of immigrant women. Future lines of research could aim to analyse the differences in this regard. Finally, the countries of origin of the immigrant PEH are not known, so it is not known whether drug use is higher in PEH from some countries of origin than from others. It may be that in some countries some drugs are more normalised than others, as is the case with alcohol in Spain. It would be interesting to investigate whether immigrants’ drug use adapts to the culture of their destination country or whether they continue to consume substances that are more culturally accepted in their countries of origin.

## 5. Conclusions

In conclusion, there are no significant differences in the drug use of men and women experiencing homelessness. To investigate the issue further, it would be necessary to investigate the type of drugs used by gender, to find out whether the results agree with some studies that show that women use more legal drugs and men use more illegal drugs. In recent years, an increasing number of women have been using legal substances (tobacco, alcohol, and psychotropic drugs) [76]. To a certain extent, the type of drug denotes the reason it is used, which can be for recreational use or self-medication, among others. A high percentage of homeless women are known to have experienced traumatic experiences and severe and violent aggression [29], so it is not surprising that they use psychotropic drugs. Illegal drug use among women carries with it guilt, fear of the public, and an unfavourable emotional burden, due to the beliefs imposed by the patriarchal system that extols the model of a woman dependent on her husband, and of the caring mother [86]. Drug use depends on which illegal drug is not seen as feminine and what is expected of a woman [87,88]. The gender perspective is fundamental for making inequalities visible; inequalities that are understood as being in continuous change, under construction, and reproduced through everyday interactions [89]. The gender system categorises, stereotypes, orders, and assigns values and norms, fostering risk factors based on the differences in exposure and vulnerability between men and women [8].

Following the Spanish Government’s failure to implement the Comprehensive National Strategy for the Homeless (*Estrategia Nacional Integral para Personas Sin Hogar*) approved in 2015, with the degree of implementation barely reaching 38% by January 2020, strong pressure was exerted by third sector organisations, with the result that the following actions were finally carried out in 2022:

The Council of Ministers have approved a Law to address the social emergency in the area of housing [90], which defines and addresses the problem of homelessness for the first time. Different parliamentary groups have registered proposals for the future Law to focus on the eradication of this phenomenon through solutions based on housing.

The State Plan for Access to Housing 2022–2025 [91] has been approved, which includes an aid programme specifically aimed at facilitating access to housing for the homeless. Several autonomous regions have already launched this programme, and in 2023 it is expected to be improved and extended to more regions.

The Secretary of State for Social Rights has launched a process to design the new Homelessness Strategy 2023–2030 [92], which is expected to be more effective than the previous one. It is important that it extends beyond mere management and proposes the eradication of homelessness. Joint work and co-governance between housing and social services will be essential when carrying out an in-depth review of the care system, moving towards methodologies based on personalising support and the response in standardised housing in community settings, in line with the goal of deinstitutionalisation. Strong public leadership, accompanied by targeted and sufficient funding, will be essential to transform the system.

Thus, although progress is being made in terms of approving specific proposals for persons experiencing homelessness, it will be a few years before the effectiveness of the implemented actions can be assessed. Determining factors, such as a lack of budget, an absence of clear leadership, a lack of assessment committees to monitor the actions carried out, difficulties in designing the plan itself, as well as in defining and delimiting the problem of homelessness, which is aggravated by the growth of other adjacent phenomena, such as residential exclusion, make it very arduous and laborious to carry out projects and future lines of work, such as those proposed here, which no longer address the needs of the homeless in general, but of those most vulnerable, such as women and immigrants [93].

Further research is, therefore, needed into why men and women experiencing homelessness use drugs. Proposals for interventions from a gender perspective that address the reasons why PEH use drugs are also necessary. Although there are no significant differences in the level of drug use in men and women, there may be in the causes of drug use and the type of drugs used, leading to different kinds of dependence.

Immigrant PEH have a lower risk of drug use than Spanish PEH. This may be related to the reasons they are homeless, which may be more because of structural factors inherent to the process of immigration—the lack of documentation, unemployment, or the absence of links to the community network—and not to individual factors, such as separation from their families or drug use prior to immigration. Drug users probably do not migrate. Being homeless is a risk factor that can trigger drug use and risky and dependent behaviour. It is, therefore, necessary to implement better reception plans for immigrants that help maintain their health and avoid homelessness. These plans should address all types of issues related to the requirements of the destination country’s system, from socio-educational and cultural programmes to integration programmes—even for those without documentation, with the aim of obtaining it through work—to the provision of social housing that ensures that basic needs are met.

## Figures and Tables

**Table 1 ijerph-20-04007-t001:** Sampling inclusion criteria according to the ETHOS classification.

Operational Category		Living Situation		Definition
1	People living rough	1.1	Public space or external space	Living in the streets or public spaces, without a shelter that can be defined as living quarters
2	People in emergency accommodation	2.1	Night shelter	People with no usual place of residence who make use of overnight shelter or low threshold shelter
3	People living in accommodation for the homeless	3.1	Homeless hostel	Where the period of stay is intended to be short term
		3.2	Temporary accommodation	
		3.3	Transitional supported accommodation	

**Table 2 ijerph-20-04007-t002:** Sociodemographic characteristics of the sample of PEH.

Variable	Value
Age ^M (SD)^	51.2 (11.4)
Gender ^n (%)^	
Man	187 (57.5)
Woman	130 (40.0)
Transexual	6 (1.8)
Nationality ^n (%)^	
Spanish	170 (52.3)
Immigrant	132 (40.6)
Dual	21 (6.5)
Civil status ^n (%)^	
Single	201 (61.8)
Married ^a^	25 (7.7)
Separated ^b^	88 (27)
Widow	11 (3.4)
Sexual orientation ^n (%)^	
Heterosexual	298 (91.7)
Bisexual	12 (3.7)
Homosexual	10 (3.1)
Other	3 (0.9)
Education ^b n (%)^	
No education	17 (5.2)
Incomplete primary education	28 (8.6)
School leaving certificate	107 (32.9)
Secondary school ^c^	120 (37.0)
University ^d^	52 (16.0)

^M^ Median. ^SD^ Standard Deviation. ^n^ Total. ^%^ Percentage. ^a^ Includes civil partnerships. ^b^ Includes separated and divorced married couples and separated civil partnerships. ^c^ Includes secondary and higher non-university studies (vocational training). ^d^ Including university studies, higher education, and doctorates.

**Table 3 ijerph-20-04007-t003:** Results of the instruments for measuring drug use.

Variable	Value
DAST-10 scores ^M (SD)^	1.26 (2.64)
Low risk or abstinence ^n (%)^	228 (70.2)
Risky use ^n (%)^	10 (3.1)
Harmful use ^n (%)^	26 (8.0)
Dependence ^n (%)^	32 (9.8)
Grouped DAST-10 scores	
Non-harmful use ^n (%) a^	238 (73.2)
Harmful use ^n (%) b^	58 (17.8)
SDS score ^M (SD)^	5.34 (4.86)
No dependence ^n (%)^	35 (10.8)
Dependence ^n (%)^	43 (13.2)
AUDIT score ^M (SD)^	12.37 (9.11)
No-risk use ^n (%)^	31 (9.5)
Risky use ^n (%)^	57 (17.5)
Dependence ^n (%)^	47 (14.5)
Scales grouped together ^c^	
Harmful or risky use	105 (32.3)
Dependence	86 (26.5)

^M^ Median. ^SD^ Standard Deviation. ^n^ Total. ^%^ Percentage. ^a^ For DAST-10, non-harmful use is low-risk use or abstinence plus risky use. ^b^ For DAST-10, harmful use is harmful use plus dependence. ^c^ Unification of scores for harmful use, risky use, and/or dependence on any substance with all the measuring instruments. DAST-10: Drug Abuse Screening Test; SDS: Severity Dependence Scale; AUDIT: Alcohol Use Disorders Identification Test.

**Table 4 ijerph-20-04007-t004:** Gender differences among PEH according to instruments for the assessment of drug and alcohol abuse and dependence (DAST-10-10, AUDIT, SDS).

Variable	Gender *		Values			
	Men	Women	x^2^/U	gl	*p*	V de Cramer
DAST-10 scores ^M (SD)^	1.13 (2.49)	1.04 (2.45)	7735.0	248	0.966	-
Low risk or abstinence ^n (%)^	109 (80.1)	91 (79.8)	2.178	2	0.563	0.093
Risky use ^n (%)^	28 (1.5)	5 (4.4)				
Harmful use ^n (%)^	12 (8.8)	8 (7.0)				
Dependence ^n (%)^	13 (9.6)	10 (8.8)	0.046	1	0.830	0.014
Grouped DAST-10 scores						
Non-harmful use ^n (%) a^	111 (81.6)	96 (84.2)	0.293	1	0.588	0.034
Harmful use ^n (%) b^	25 (18.4)	18 (15.8)				
SDS score ^M (SD)^	5.74 (5.09)	5.22 (5.23)	340.0	53	0.668	-
No dependence ^n (%)^	13 (48.1)	12 (50.0)	0.017	1	0.895	0.018
Dependence ^n (%)^	14 (51.9)	12 (50.0)				
AUDIT score ^M (SD)^	11.47 (10.02)	14.73 (11.02)	1059.5	65	0.225	-
No-risk use ^n (%)^	14 (41.2)	11 (33.3)	0.440	1	0.507	0.081
Risky use ^n (%)^	20 (58.8)	22 (66.7)				
Dependence ^n (%)^	17 (50.0)	21 (63.6)	1.268	1	0.026	0.138
Scales grouped together ^c^						
Harmful or risky use	40 (29.0)	35 (28.9)	<0.001	1	0.992	0.001
Dependence	31 (22.5)	32 (26.4)	0.555	1	0.456	0.046

^M^ Median. ^SD^ Standard Deviation. ^n^ Total. ^%^ Percentage. * The six transgender women are not included and the two people who did not want to reveal their gender are not included. ^a^ For DAST-10, non-harmful use is low-risk use or abstinence plus risky use. ^b^ For DAST-10, harmful use is harmful use plus dependence. ^c^ Unification of scores for harmful use, risky use, and/or dependence on any substance with all the measuring instruments. DAST-10: Drug Abuse Screening Test, SDS: Severity Dependence Scale, AUDIT: Alcohol Use Disorders Identification Test.

**Table 5 ijerph-20-04007-t005:** Differences by origin among homeless people according to instruments for the assessment of drug and alcohol abuse and dependence (DAST-10, SDS, AUDIT).

Variable	Nationality		Values			
	Spanish	Immigrant	x^2^/U	gl	*p*	V de Cramer
DAST-10 scores ^M (SD)^	1.63 (3.05)	0.95 (2.15)	8394.0	272	0.088	-
Low risk or abstinence ^n (%)^	113 (72.4)	96 (80.7)	5.063	2	0.167	0.136
Risky use ^n (%)^	6 (3.8)	4 (3.4)				
Harmful use ^n (%)^	13 (8.3)	11 (9.2)				
Dependence ^n (%)^	24 (15.4)	8 (6.7)	4.926	1	0.026	0.134
Grouped DAST-10 scores						
Non-harmful use ^n (%) a^	119 (76.3)	100 (84.0)	2.501	1	0.114	0.095
Harmful use ^n (%) b^	37 (23.7)	19 (16.0)				
SDS score ^M (SD)^	6.00 (5.19)	4.66 (4.22)	620.0	77	0.279	-
No dependence ^n (%)^	2 (44.0)	10 (40.0)	0.109	1	0.714	0.038
Dependence ^n (%)^	28 (56.0)	15 (60.0)				
AUDIT score ^M (SD)^	13.36 (10.39)	9.61 (7.79)	590.5	82	0.265	-
No-risk use ^n (%)^	20 (32.8)	9 (39.1)	0.297	1	0.586	0.059
Risky use ^n (%)^	41 (67.2)	14 (60.9)				
Dependence ^n (%)^	34 (55.7)	11 (47.8)	0.420	1	0.057	0.071
Scales grouped together ^c^						
Harmful or risky use	72 (42.4)	29 (22.0)	13.869	1	<0.001	0.214
Dependence	60 (35.3)	24 (18.2)	10.837	1	0.001	0.189

^M^ Median. ^SD^ Standard Deviation. ^n^ Total. ^%^ Percentage. ^a^ For DAST-10, non-harmful use is low-risk use or abstinence plus risky use. ^b^ For DAST-10, harmful use is harmful use plus dependence. ^c^ Unification of scores for harmful use, risky use, and/or dependence on any substance with all the measuring instruments. DAST-10: Drug Abuse Screening Test, SDS: Severity Dependence Scale, AUDIT: Alcohol Use Disorders Identification Test.

## Data Availability

The data presented in this study are available on request from the corresponding author. The data are not publicly available due to ethics and privacy requirements.

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
