# Peer review of "Differences in Drug Use among Persons Experiencing Homelessness According to Gender and Nationality"

_ijerph, 2023, doi:10.3390/ijerph20054007_

Round 1
Reviewer 1 Report
Review: “Differences in drug use among persons experiencing homelessness according to gender and nationality”.
The paper addresses an important and relevant issue both for the academic literature and for public policies on the subject: drug and alcohol consumption in the homeless population. The findings also seem interesting because they find that alcohol consumption among women and men is relatively similar and also because they find that Spaniards have higher consumption than the migrant population (i.e., national culture is a determinant of such consumption). Having said this, the paper presents a series of problems that need to be worked on before publication.
The introduction is too vague and needs to be more precise: problem, literature gap, key research question, methods and main findings. But perhaps the most important thing in relation to this part of the paper is that the literature presented in the introduction is insufficient: it is necessary to develop a complete section framing the discussion of these issues. This frame should point out the gaps that the paper intends to fill with new data and analysis. On the other hand, it serves to frame the findings in the discussion of the topic in order to correctly determine their value.
However, beyond these drawbacks, I believe that the main objection is perhaps the representativeness of the data and the size of the sample. With such a low N, some results become meaningless. For some variables, the N is so small that talking about a percentage or using that N in a statistical model is irrelevant because all the results usually turn out to be significant. For example, the N of women experiencing homelessness in Guadalajara is 5 (5 women), what does it mean in that situation to talk about 50% or any other percentage? Although this is a very valuable survey given the scarcity of data on the subject, it seems to me that perhaps it would have been more interesting to do qualitative work.
In the discussion, the presentation of results is merely descriptive without an in-depth analysis of their significance for the literature and for public policy. The similar use of alcohol among men and women deserves a lot of analysis and especially contrast/discussion with the literature on the subject.
In the conclusions, I believe that some deeper reflection should be included on the public policy implications of these findings and also, what could be a future line of research to advance the study of this topic.
Author Response
I am very grateful for your contributions as they have provided an opportunity to explore the issues raised a little further.
- Contribution/ Suggestion
“The introduction is too vague and needs to be more precise: problem, literature gap, key research question, methods and main findings. But perhaps the most important thing in relation to this part of the paper is that the literature presented in the introduction is insufficient: it is necessary to develop a complete section framing the discussion of these issues. This frame should point out the gaps that the paper intends to fill with new data and analysis. On the other hand, it serves to frame the findings in the discussion of the topic in order to correctly determine their value.”
- Answer
I have added literature in the Introduction, which adds information about the gaps that the research aims to fill, as well as a better explanation of the problem of stigma suffered by homeless people and especially the most vulnerable such as women and migrants. I have also better detailed the objective and key questions that motivate the study.
- Contribution/ Suggestion
“However, beyond these drawbacks, I believe that the main objection is perhaps the representativeness of the data and the size of the sample. With such a low N, some results become meaningless. For some variables, the N is so small that talking about a percentage or using that N in a statistical model is irrelevant because all the results usually turn out to be significant. For example, the N of women experiencing homelessness in Guadalajara is 5 (5 women), what does it mean in that situation to talk about 50% or any other percentage? Although this is a very valuable survey given the scarcity of data on the subject, it seems to me that perhaps it would have been more interesting to do qualitative work.”
- Answer
In the limitations section, the low representativeness of the sample is added. In addition, the possibility of carrying out a qualitative study is added as a future avenue for research.
- Contribution/ Suggestion
In the discussion, the presentation of results is merely descriptive without an in-depth analysis of their significance for the literature and for public policy. The similar use of alcohol among men and women deserves a lot of analysis and especially contrast/discussion with the literature on the subject.
- Answer
I have added a paragraph that goes a little deeper into the issue, since although there are no differences at the quantitative level, the gender perspective must be incorporated, which offers a qualitative explanation of the results. I also introduce which factors should be taken into account by public policies in the design of programmes for homeless people according to their gender.
- Contribution/ Suggestion
In the conclusions, I believe that some deeper reflection should be included on the public policy implications of these findings and also, what could be a future line of research to advance the study of this topic.
- Answer
I have added the current state of public policy on homelessness and a reflection on the difficulty of carrying out effective projects that meet the needs of homeless people, due to the rigidity of the system itself.
Reviewer 2 Report
In this research, on the population of homeless people from three Spanish cities, it was investigated whether there are gender differences or differences related to nationality in the abuse of drugs and alcohol. The subjects were interviewed, and in order to determine drug and alcohol addiction, standardized analytical instruments were used. The results did not show gender differences, but there were differences depending on nationality - immigrants had a lower prevalence of drug addiction.
The study is clearly presented: the findings of previous research are adequately presented, the subjects and the methods used are well described, the results are clear, and the conclusions are based on the results. Also, the main limitations of this research are highlighted.
I just have some minor comments/objections:
Please add a new keyword, "Spain", and change the word "migration" to "immigrants".
In the Materials & Methods section, when you indicate how many people of each gender are included in the survey, please indicate at least approximately what proportion of homeless people in each of the three cities included your subjects represent.
Given that you have collected and shown sociodemographic data in Table 2, I am interested in why you did not perform a chi-square test and compare respondents according to gender and nationality?
When describing the respondents from Girona, you stated that two people did not want to reveal their gender. I assume that these two people were not included in the analyzes shown in Table 4, so please note that in the footnote.
Why did you not include trans people in the analyzes shown in Table 5, when the differences based on nationality were investigated there? And considering that it was not pointed out in the footnote that two people from Girona who did not want to reveal their gender were excluded from this analysis, does it mean that they were included?
page 8, line 267 – there is an extra space
page 8, line 268 – please add “… between 45 and 64 years of age.”
page 8, lines 277-278 – I disagree with the statement that the proportions of men and women in your sample are similar, please rephrase that.
In the Discussion and Conclusions sections, you make hypotheses about the reasons why immigrants use drugs. During the interview, did you ask your respondents if they used drugs before coming to Spain or if they started after their arrival?
Author Response
Thank you for your precision and clarity in formulating suggestions, it has helped me to realise that I need to be more concise on certain issues.
- Contribution/ Suggestion
Please add a new keyword, "Spain", and change the word "migration" to "immigrants".
- Answer
Done.
- Contribution/ Suggestion
In the Materials & Methods section, when you indicate how many people of each gender are included in the survey, please indicate at least approximately what proportion of homeless people in each of the three cities included your subjects represent.
- Answer
Done.
- Contribution/ Suggestion
Given that you have collected and shown sociodemographic data in Table 2, I am interested in why you did not perform a chi-square test and compare respondents according to gender and nationality?
- Answer
The number of immigrant women in the sample was very small, so it was not possible to carry out the corresponding statistical analyses. Furthermore, it was considered necessary to analyse gender and immigration separately, in line with the objectives of the study. The following is included in the limitations section.
- Contribution/ Suggestion
When describing the respondents from Girona, you stated that two people did not want to reveal their gender. I assume that these two people were not included in the analyzes shown in Table 4, so please note that in the footnote.
- Answer
True, two people who did not want to disclose their gender were not included. I have added the footnote.
- Contribution/ Suggestion
Why did you not include trans people in the analyzes shown in Table 5, when the differences based on nationality were investigated there? And considering that it was not pointed out in the footnote that two people from Girona who did not want to reveal their gender were excluded from this analysis, does it mean that they were included?
- Answer
I have checked the data and transgender people were included, it was a mistake to write the footnote in Table 5. I have removed it.
On the other hand, people who did not state their gender are also included.
- Contribution/ Suggestion
page 8, line 267 – there is an extra space
- Answer
Done.
- Contribution/ Suggestion
page 8, line 268 – please add “… between 45 and 64 years of age.”
- Answer
Done.
- Contribution/ Suggestion
page 8, lines 277-278 – I disagree with the statement that the proportions of men and women in your sample are similar, please rephrase that.
- Answer
Done.
- Contribution/ Suggestion
In the Discussion and Conclusions sections, you make hypotheses about the reasons why immigrants use drugs. During the interview, did you ask your respondents if they used drugs before coming to Spain or if they started after their arrival?
- Answer
In the interviews I did not ask direct questions about this issue, but there were conversations that tended to state that they started using drugs as a result of becoming homeless, which seemed logical but also relevant and to be considered in the analysis of the reason for their drug use.
Round 2
Reviewer 1 Report
The authors have incorporated most of the suggestions and comments. The text has been improved in the presentation of results and content analysis.